**DOI: 10.1038/ncomms13687**　　**OPEN**

# Lasing in dark and bright modes of a finite-sized plasmonic lattice

T.K. Hakala[1,*], H.T. Rekola[1,*], A.I. Väkeväinen[1], J.-P. Martikainen[1], M. Nečada[1], A.J. Moilanen[1] & P. Törmä[1]

Lasing at the nanometre scale promises strong light-matter interactions and ultrafast operation. Plasmonic resonances supported by metallic nanoparticles have extremely small mode volumes and high field enhancements, making them an ideal platform for studying nanoscale lasing. At visible frequencies, however, the applicability of plasmon resonances is limited due to strong ohmic and radiative losses. Intriguingly, plasmonic nanoparticle arrays support non-radiative dark modes that offer longer life-times but are inaccessible to far-field radiation. Here, we show lasing both in dark and bright modes of an array of silver nano-particles combined with optically pumped dye molecules. Linewidths of 0.2 nm at visible wavelengths and room temperature are observed. Access to the dark modes is provided by a coherent out-coupling mechanism based on the finite size of the array. The results open a route to utilize all modes of plasmonic lattices, also the high-$Q$ ones, for studies of strong light-matter interactions, condensation and photon fluids.

[1] COMP Centre of Excellence, Department of Applied Physics, Aalto University School of Science, FI-00076 Aalto, Finland. * These authors contributed equally to this work. Correspondence and requests for materials should be addressed to P.T. (email: paivi.torma@aalto.fi).

Plasmonic systems offer small mode volumes, ultrafast dynamics and nanoscale operation, which opens new prospects for light-matter interactions. Strong coupling between plasmonic modes and ensembles of emitters has been demonstrated[1,2], and strong coupling even at the level of a few emitters has been reached at room temperature[3,4]. Gain assisted propagation of plasmon polaritons, spacing and lasing have been predicted[5–9] and studied experimentally[10–24], and the possibility of photon condensates has been proposed[25]. The fundamental tradeoff between the confinement of optical fields and losses[26] render plasmonic systems inherently lossy. This motivates a search for hybrid modes where losses can be reduced by a narrow-linewidth component while still preserving the near-field characteristics of the plasmonic component.

Lattices of metal nanoparticles support modes called surface lattice resonances (SLRs) which are hybrids of localized nanoparticle surface plasmon resonances and diffracted orders (DOs) of the periodic structure[27–30]. The mode energies, losses and optical density of states can be tuned by lattice and particle geometry. The SLRs combine extreme optical confinement with the narrow linewidths inherent in the DO part of the hybrid. Indeed, strong coupling with organic molecules in such a system has been observed[2,31,32]. Infrared lasing at weak coupling regime has been realized in such lattices[15,24], with linewidths of <1.3 nm (ref. 15). As the losses of an optical dipole scale as $\omega^4$, noble metals can suffer considerably higher losses at visible compared to IR. A key question has been whether similar nanoparticle structures could support lasing also at visible frequencies.

Importantly, plasmonic lattices can also support so-called dark SLR modes, whose subradiant character results to significantly higher Q-values as compared with their radiant (bright) counterparts[33]. Thus plasmonic dark modes are promising candidates for realizing lasing, single-emitter strong coupling, and photon fluids at visible wavelengths. A central challenge is how to excite and couple out these inherently subradiant modes.

Here, we experimentally demonstrate lasing at the visible wavelengths in both bright and dark modes of the plasmonic lattice. A new concept to access the dark modes is introduced, which is based on a gradual, coherent build-up of dipole moments in a finite lattice.

## Results

**Dark and bright modes of an infinite lattice.** We fabricate and measure arrays of silver nanoparticles on glass, with interparticle spacings of 360–400 nm and array sizes of $100 \times 100 (\mu m)^2$, see Methods. Figure 1a–c shows the measurement setup, schematic and scanning electron micrograph of a sample. In Fig. 1d the measured dispersion $E(k_x)$ of a typical sample is shown, where $k_x$ is the in-plane momentum in the x-direction. Yellow features correspond to the bright modes that respond to the far-field radiation incident on the sample in the transmission measurement, while the location of the dark mode is indicated by a circle. Figure 1e,f shows the corresponding electric field and charge distributions found by Finite-Difference Time-Domain (FDTD) simulations for an infinite array. In Fig. 1e, a standing wave antinode at each particle location induces a large dipole moment

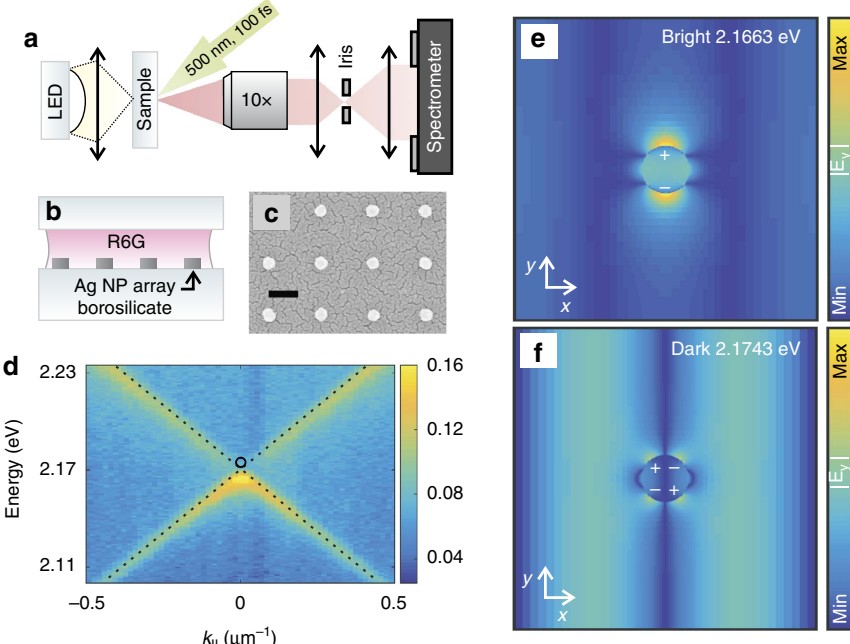

**Figure 1 | Measured dispersion of the SLR modes. (a)** The angle-resolved transmission/luminescence spectra were collected by focusing the image of the backfocal plane of the objective to the entrance slit of the spectrometer. For transmission measurements, a white light source was used. For lasing measurements, the gain medium was pumped with a femtosecond pulsed laser. **(b)** A schematic of the sample, consisting of rectangular arrays of silver nanoparticles on a borosilicate substrate, a gain medium and a cover glass. **(c)** A scanning electron micrograph of a typical sample. The scale bar is 200 nm. **(d)** Measured extinction of a typical sample, showing the dispersion of the SLR, which results from the hybridization of the nanoparticle surface plasmon resonance with the diffracted $\langle +1,0 \rangle$ and $\langle -1,0 \rangle$ orders of the lattice (denoted by dashed lines; the crossing of the dashed lines is referred to as the $\Gamma$-point). The location of the dark mode is indicated by the black circle. The colour bar indicates the value of extinction (1-transmission). **(e–f)** Charge and field distributions of the bright and dark modes for the infinite lattice, obtained from Finite-Difference Time-Domain (FDTD) simulations with periodic boundary conditions. The character of these modes can be understood by considering an array of nanoparticles with period $p_x = p_y$ mainly polarized along y direction and thus radiating predominantly along the $-x$ and $+x$ direction. Two counter-propagating radiation fields result to a standing wave which has either a node or an antinode at each particle location, corresponding to the dark or bright mode at $k=0$, respectively.

and radiation to the far field, thus it is referred to as a bright mode. In Fig. 1f, however, the field gradient related to a field node induces a quadrupole moment into each particle resulting to zero net dipole moment and negligible far-field radiation, a feature of a dark mode. For both cases, the plasmonic component of the hybrid is evident from the strong near fields in the particle vicinity. A dark mode here specifically refers to one band-edge mode missing in the far field. There are other categories of dark modes in plasmonics systems which present a subradiant mode, especially in metamaterials with symmetry-breaking features[34,35].

**Lasing.** We combine the lattices with a 31 mM solution of Rhodamine 6G molecules. For concentrations of this order of magnitude, formation of aggregates is not significant[36]. Addition of the dye molecules modifies the refractive index surrounding the structure, shifting the SLR resonances slightly from Fig. 1. We excite the molecules with a 100 fs laser pulse (500 nm centre wavelength) and observe the emission, see Methods. Figure 2a–c shows the momentum and energy distribution of the sample emission with different pump fluencies. Below threshold, the emission closely follows the dispersion of the SLR mode (Fig. 2a).

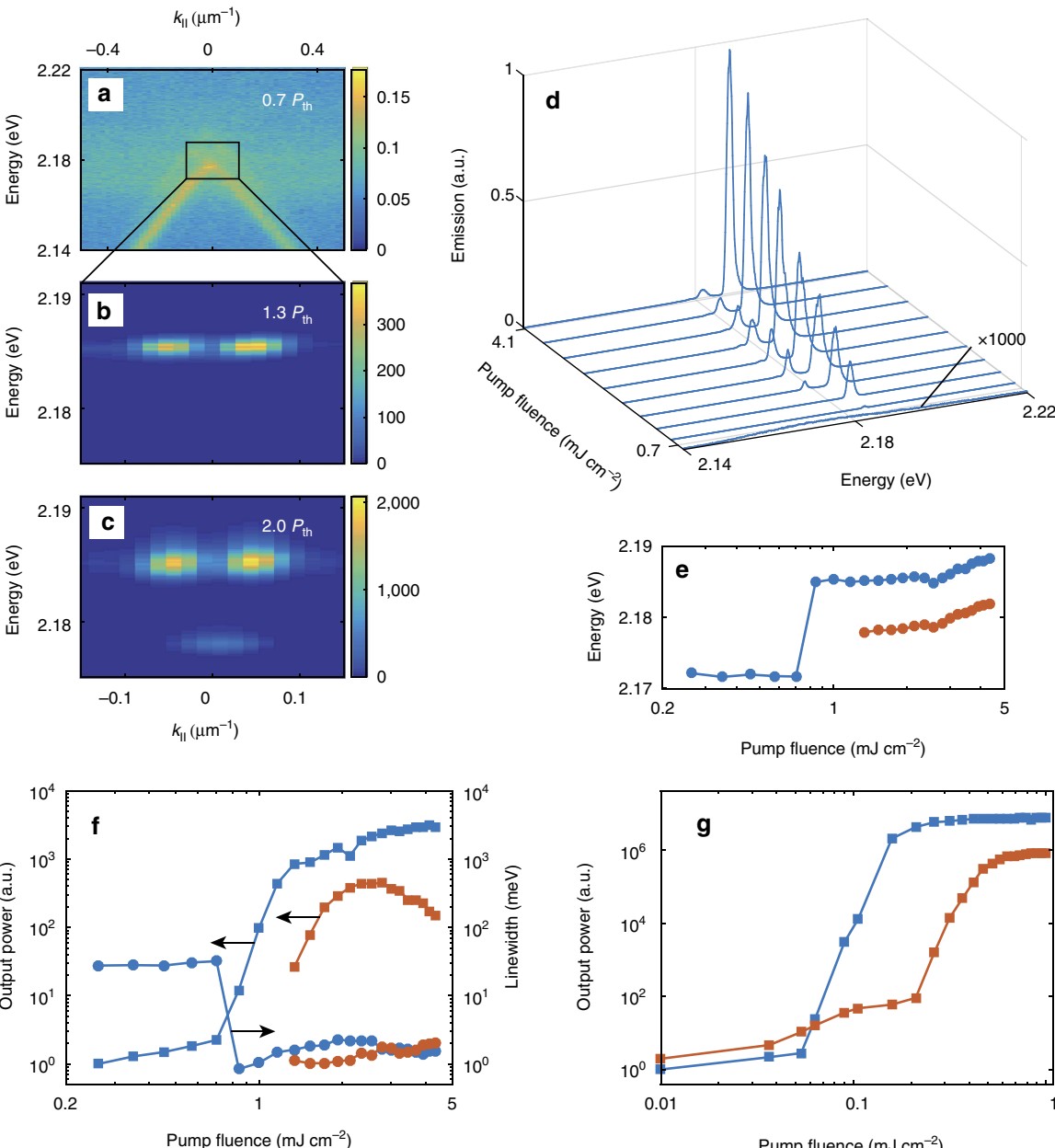

**Figure 2 | Light emission below and above the lasing threshold.** Measured emission intensities of a $100 \times 100 \, (\mu m)^2$ nanoparticle array (array periods $p_x = p_y = 375$ nm, particle diameter $d = 60$ nm) with (**a**) $P = 0.7 P_{th}$, (**b**) $P = 1.3 P_{th}$ and (**c**) $P = 2.0 P_{th}$, where $P_{th}$ is the threshold pump fluence for the higher energy lasing mode. In **a**–**c**, the colour bars indicate the emission intensity in arbitrary units. (**d**) Emission spectra, (**e**) the mode energies and (**f**) the mode output powers of the lower (red squares) and higher (blue squares) energy modes as a function of pump fluence. Also shown are the linewidths for lower (higher) energy modes as red (blue) dots. (**g**) FDTD simulation with the gain medium described as four-level systems, the red (blue) squares corresponding to the lower (higher) energy mode, respectively. We have observed lasing for array periodicities 370–390 nm. Above threshold, we obtain $10^4$ increase in emission intensity, linewidths of 0.2 nm and remarkably low beam divergence of $\Delta\theta_{FWHM} = 0.3°$.

The emission spectra in Fig. 2d shows that at threshold, an intense and narrow emission peak at 2.185 eV is observed, and well above threshold a second peak appears at 2.178 eV. The momentum distributions of these two peaks are shown in Fig. 2b,c, respectively. For both peaks, characteristic signatures of lasing, such as rapid nonlinear increase of emission intensity, reduction of linewidth, and a blue shift of the emission are observed in response to increasing pump fluence, see Fig. 2e,f. Further, a high beam directionality with divergence of 0.3° is observed for both modes. The qualitative behaviour of lasing thresholds is well reproduced by an FDTD simulation with the dye molecules described as four-level systems (for details see Methods), as shown by Fig. 2g. The simulations show lasing at two separate energies, 2.169 and 2.142 eV. The higher energy mode has a lower lasing threshold, analogous to the experiments.

**Control experiments**. To further characterize the lasing, several control experiments were carried out. First, a sample with equal area and amount of nanoparticles but at random positions exhibits no lasing, ruling out random lasing[37]. FDTD simulations of our structures show that the near fields decay, in the $z$-direction perpendicular to the array, to 5% of the peak value within 100 nm. We deposited polymer poly(vinyl alcohol) (PVA) layers of various thicknesses on top of the array so that the gain

medium was separated from the nanoparticles: for thickness of about 100 nm we observed lasing, but not for 500 nm (Supplementary Fig. 1; Supplementary Note 1). For phenomena reliant on strong coupling, the presence of the emitters extremely close to the nanoparticles is expected to be more essential than for our lasing which takes place in the weak coupling regime. As one more control, in contrast to ref. 23 where SLRs hybridize with waveguide modes, our samples exhibit no lasing when over 75% of particles are removed. For more information about the control experiments with 0, 50 and 75% of particles removed, see Supplementary Figs 2–4 and Supplementary Note 1.

**Real space images**. From Fig. 2d–f we note that the higher energy lasing mode has a lower threshold, higher emission intensity, and a slightly narrower linewidth compared with the lower energy peak. It could therefore be associated with the low loss dark mode. However, in case of an infinite array size, the dark mode should not radiate to the far field, see Fig. 1f. Curiously, the higher energy mode exhibits multiple peaks in $k$, whereas the lower energy lasing mode shows only a single peak at $k = 0$, see Fig. 2c. A striking difference between the two cases is observed also in real space images of the lasing action. These far field microscope images correspond to the real space distribution of radiation from the sample but are not the same as the near-field

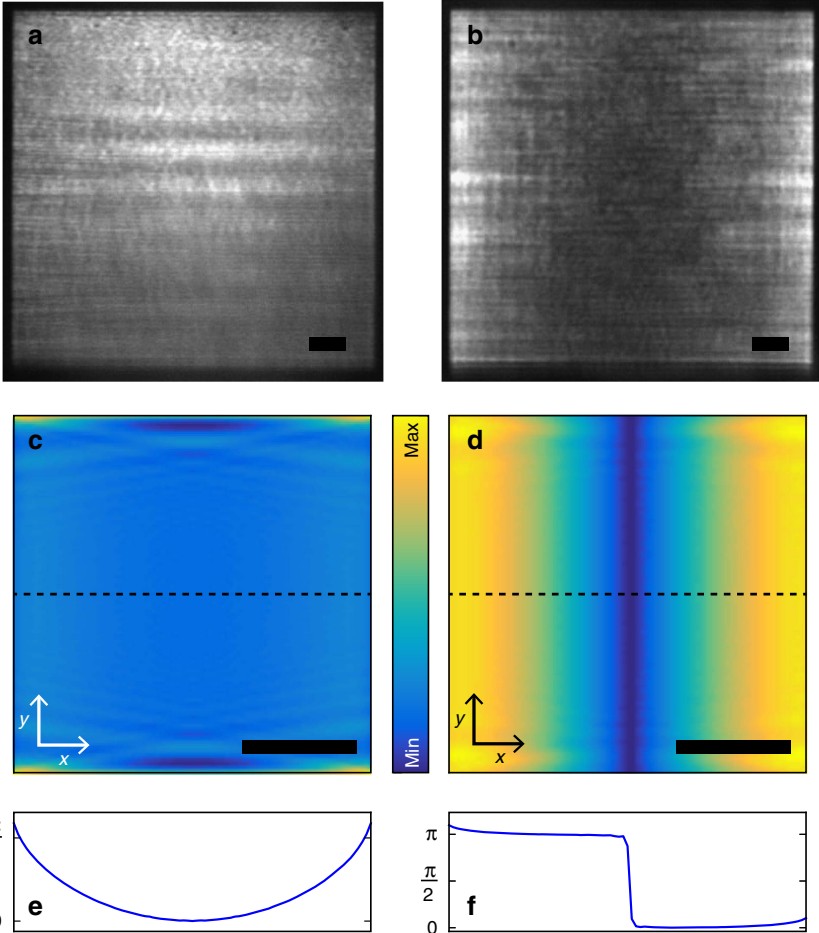

**Figure 3 | Real space images of the lasing action and multipole expansion of scattering for *y*-polarized planewave.** (**a–b**) Examples of experimentally observed real space images of the lower (**a**) and upper (**b**) energy lasing modes from two different samples. (**c–d**) Magnitude of the dipole moments (indicated by the colour bar) as a function of position in an array of 85 × 85 silver nanoparticles, computed using a multipole expansion method without a gain medium[43], at lower (**c**) and higher (**d**) mode energies. The calculated phase of the dipole moments along the dashed lines for both modes are shown in **e,f**. The scale bars in **a–d** are 10 μm.

distribution. Figure 3a,b shows that the lower energy mode lases in the middle of the array, while in the case of the upper energy mode the lasing light is predominantly visible at the edges. In Fig. 3c,d are shown multipolar scattering simulations, without a gain medium, of a finite $85 \times 85$ particle array under the $y$-polarized plane wave excitation (for details see Methods). In the simulations the particle polarizabilities include both dipolar and quadrupolar components. The real-space images in far field result from radiation emitted by the sample. The far field emission power is dominated by dipoles, thus the radiated far field intensities in Fig. 3a,b are proportional to the square of induced dipole moments. Notably, the phase of the dipole moment stays nearly constant across the array for the lower energy mode (Fig. 3e), whereas for the higher energy mode the phase undergoes an abrupt $\pi$-phase shift at the centre of the array (Fig. 3f). The amplitudes and phases of the quadrupolar moments for both modes are shown in Supplementary Fig. 5, and the related discussion can be found from Supplementary Note 2.

**Finite size effects**. We hypothesize that the higher energy mode is indeed the dark mode, and that its visibility, the peculiar beating pattern in $k$ and the real space distribution of the lasing light are all consequences of gradual evolution of radiation fields in the array due to finite-size effects. Figure 4 shows schematically how finite size effects can lead to radiating dipoles near the edge of the array that are either in the same (bright mode) or opposite (dark mode) phase. Importantly, such a picture holds only if spatial coherence is preserved over the whole array. The profoundly different character of the two modes is manifested in their dipole moment distributions. According to the argument of Fig. 4, for the bright (dark) mode, the induced dipole moments maximize in the centre (edges) of the array. The dark mode in a finite lattice is thus a hybrid mode containing also features typical for a bright mode of an infinite lattice, such as dipoles, which makes it visible in the far field. The expected radiation in real space is strongest where the dipoles maximize. Indeed, in the real space images of the lasing action, the bright mode mainly emits from the centre of the array (Fig. 3a), whereas the dark mode radiates from the edges (Fig. 3b). Similar behaviour is visible in the simulations (Fig. 3c,d).

The different real-space distributions of the two lasing modes are expected to lead to distinct behaviour of the corresponding far-field momentum distribution. Analogous to single slit interference, the bright mode acts as a source with a constant phase and nearly equal amplitude, leading to an odd beating pattern in $k$ (constructive interference at $k = 0$). For the dark mode, radiation maximizes towards the edges, and thus a double slit analogy can be made. Due to the $\pi$-phase shift in the radiation fields across the array, an even beating pattern in $k$ is expected (destructive interference at $k = 0$). Indeed, in Fig. 2c, we observe a peak at $k = 0$ for the lower energy mode and two peaks shifted away from $k = 0$ for the higher energy, as expected for bright and dark modes, respectively.

As the ultimate test to our hypothesis, the even beating pattern in $k$ for the dark mode should depend on the array size. This is exactly what we observe: as the array size is decreased, the spacing of the beating pattern maxima gradually increases (Fig. 5), depending linearly on the inverse of the array size, which is expected for small angles. The associated beating pattern in $k$ shows that spatial coherence of the lasing action extends over the whole array for the system sizes (50–100 m) considered. From the lasing spectra for different array sizes (Fig. 5c) it is apparent that while smaller arrays have a higher pump threshold, the mode linewidth is not increased significantly. We have thus shown lasing in the dark mode which

can be coupled out by a unique approach based on the finite size of the array.

## Discussion

In summary, we have demonstrated lasing at visible wavelengths both in dark and bright modes of a nanoparticle array. Even though plasmonic systems at visible wavelengths are lossy, we achieve remarkably narrow (0.2 nm) linewidths, an increase of four orders of magnitude in emission intensity above threshold, low beam divergence of 0.3°, and spatial coherence lengths of 100 μm. This achievement, together with earlier reaching of the strong coupling regime[31,32], paves the way for studies of phenomena, which combine strong interactions with macroscopic mode population, for example thresholdless lasing, photon and polariton condensation, and photon fluids. The dark-mode out-coupling mechanism that we introduce is not simple scattering or leakage from sharp edges of the system, rather, it is gradual, coherent build-up of dipole moments and radiation intensity. This inspires ideas for the design of not only out-coupling schemes but also beam guiding, trap potentials, topologically non-trivial lattices and edge modes, for instance by gradually changing the pitch and by particle shapes supporting higher order multipoles.

## Methods

**Sample fabrication.** Using electron beam lithography, rectangular arrays of cylindrical (nomimal diameter 60 nm, height 30 nm) silver nanoparticles were fabricated on a borosilicate substrate. The angle resolved transmission spectra were obtained by focusing light from a white LED onto a sample. A $10 \times 0.3$ NA objective focused onto the sample surface collected the transmitted light.

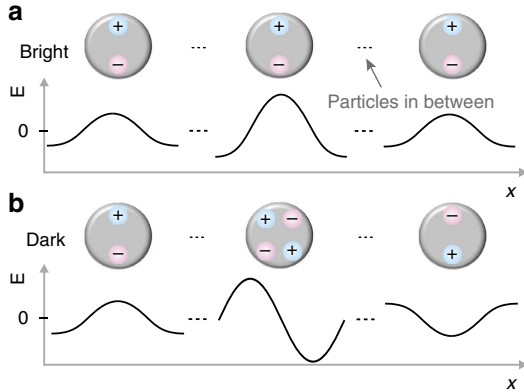

**Figure 4 | Evolution of the radiation fields in the case of a finite array size.** (**a**) For a bright mode, the phase across the array remains constant due to constructive interference of the counter-propagating radiation fields at each particle location. The amplitude at the edge, however, will be reduced to one half as there are no particles and therefore no radiation incident from the other side. (**b**) For a dark mode, particles in the array are driven by left- and right-propagating radiation fields, which destructively interfere at the particle locations and therefore create a standing wave node and a quadrupole excitation. As one moves away from the centre; however, the destructive interference becomes gradually less complete due to unequal number of particles contributing to left- and right-propagating waves. This results in a buildup of dipole moments whose relative weight compared to the quadrupoles gradually increases when moving away from the centre of the array. Importantly, for the dark mode the dipoles induced by the left- and right-propagating radiation fields are out of phase by $\pi$. The dark mode in a finite lattice is thus different from the one in an infinite lattice: it features dipole moments and can thus be viewed as a hybrid mode with characteristics of both the bright and dark modes of an infinite lattice.

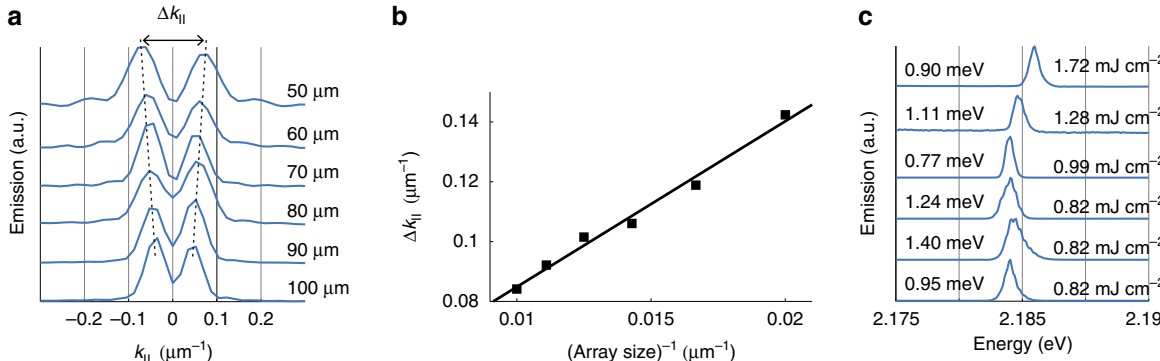

**Figure 5 | Dependence of the k-space beating pattern on the array size.** (**a**) Normalized constant energy crosscut of the dark mode lasing peak as a function of in-plane k-vector for different array sizes. (**b**) The separation of the peaks in **a** as a function of 1/(array size). The separation of the lasing peaks for 100 and 50 μm array corresponds to angular separation of 0.44° and 0.74°, respectively. Divergence of individual peaks vary between 0.29° and 0.44°. (**c**) The lasing spectra for each array size integrated over the double peak from $k = -0.2$ to $k = 0.2\,\mu m^{-1}$. The threshold fluence $P_{th}$ depends on the array size. The linewidth of the lasing peak is denoted on the left and $P_{th}$ on the right.

**Measurement setup.** The image of the back focal plane of the objective was focused to the entrance slit of the spectrometer. Thus the transmitted (or emitted) light from the sample into a given angle is focused into a single position on the entrance slit. One axis of the two-dimensional (2D) CCD camera of the spectrometer can then be used to resolve the angle of the transmitted/emitted light, while the other axis is used to resolve the wavelength. The $E(k)$ dispersions were subsequently calculated from the angle and wavelength resolved spectra as $E = hc/\lambda$ and $k_{||} = k_0 \sin(\theta)$. Here $k_0 = 2\pi/\lambda$ and $\theta$ is the angle with respect to sample normal. The real space images were obtained by focusing the image plane of the same sample onto a separate 2-D CCD camera.

For lasing measurements, the gain medium was pumped with a femtosecond laser (45° incident angle, 500 nm centre wavelength, 100 fs pulse width, 1 kHz repetition rate). The gain medium consisted of 31 mM Rhodamine 6G in Benzyl Alcohol.

**Data analysis.** The beam divergence is obtained from the angle resolved lasing data (Fig. 2) by plotting the emission intensity as a function of in-plane k-vector at the lasing energy. From this crosscut, we can determine the full-width-half-maximum value of each individual lasing peak, $\Delta k_{FWHM}$. For small angles $\sin(\theta) \approx \theta$, and thus the divergence can finally be defined as $\Delta\theta_{FWHM} = \Delta k_{FWHM}/k_0$ (rad), where $k_0$ is the free space wavenumber. The angular separation of the two peaks at dark mode energy (Fig. 5) can be calculated with the previous equation by replacing $\Delta k_{FWHM}$ with peak separation $\Delta k$.

**Simulations.** FDTD simulations were done using Lumerical's FDTD Solutions software. The nanoparticle was modelled as a 30 nm tall cylinder with a diameter of 60 nm. Tabulated material parameters for silver were used[38]. The background refractive index was set to 1.52. For the field profile simulations the simulation mesh was set to 0.6 nm over the volume of the nanoparticle. Ten dipole sources with random orientation, location, and phase excite the SLR modes with a 3.3 fs pulse at the beginning of the simulation. Field profiles were recorded after the simulation had ran for 700 fs to filter out the excitation pulse. An infinite array was considered in the simulations by choosing the $x$ and $y$ boundary conditions to be either asymmetric and symmetric (bright mode, Fig. 1e) or asymmetric and asymmetric (dark mode, Fig. 1f), respectively. This selection of boundary conditions allows to isolate the modes of interest, and control the polarization direction of the bright mode. As the structure has the same periodicity in the $x$ and $y$, equivalent modes exist also in the perpendicular direction to the one shown in Fig. 1e,f.

The simulations with gain in FDTD were done using a similar model, except this time periodic boundary conditions were used for both $x$ and $y$ to allow both the bright and dark modes to exist simultaneously. The excitation of the modes is done by pumping the dye molecules with a 500 nm, 30 fs, 0.01–2.5 mJ cm$^{-2}$ pulse normally incident on the structure. The dye is modelled as four-level systems, which are coupled to the electric fields in time domain, similar to ref. 39. After the pump pulse, the excitations in the dye molecules relax to the upper lasing state, and from there excite the bright and dark SLR modes through stimulated emission if the population inversion is sufficient to overcome the losses in the system, see Supplementary Figs 6 and 7 and Supplementary Note 3. For the parameters used to model the dye and for a more detailed description of the model, see Supplementary Fig. 8 and Supplementary Note 3. In addition to the FDTD modelling shown in the main text, we have used a neo-classical model with simplified description for the SLR modes to simulate the lasing action in a nanoparticle array[40], see Supplementary Fig. 9 and Supplementary Note 4 for details.

**Theoretical calculations.** Our theoretical calculations of the array response, see Fig. 3, are based on the multiple scattering T-matrix method[41], which is a generalization of the often-used coupled dipole approximation, enabling to take into account multipole responses of a single nanoparticle up to an arbitrary multipolar order. We compute the electromagnetic response (T-matrix) of a single nanoparticle using scuff-em, an open-source implementation of the boundary-element method (http://github.com/homerreid/scuff-EM)[42]. The nanoparticle parameters are the same as in the FDTD simulations above. All the calculations are performed up to the quadrupole order. Figure 3c shows the magnitudes of the electric dipole moment in the $y$-direction among the nanoparticles when the array is excited by the $y$-polarized plane wave coming from the normal direction, at a frequency corresponding to the bright mode. Figure 3d depicts the same $y$-dipole response when the array is excited by two $y$-polarized plane waves with opposite incident angle, corresponding to the peaks of the observed $k$-space beating patterns. A brief explanation of the multiple scattering method and figures showing the quadrupole response among the nanoparticles are provided in Supplementary Fig. 5 and Supplementary Note 2.

**Data availability.** The authors declare that relevant data supporting the findings of this study are available on request.

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

## Acknowledgements

This work was supported by the Academy of Finland through its Centres of Excellence Programme (2012–2017) and under project nos. 263347, 284621 and 272490, and by the European Research Council (ERC-2013-AdG-340748-CODE). This article is based upon work from COST Action MP1403 Nanoscale Quantum Optics, supported by COST (European Cooperation in Science and Technology). Part of the research was performed at the Micronova Nanofabrication Centre, supported by Aalto University. Triton cluster at Aalto University was used for the computations. The authors thank M. Dridi for fruitful discussions.

## Author contributions

P.T. initiated and supervised the project. T.K.H., H.T.R. and A.I.V. designed and performed the experiments and analysed the data. H.T.R. and J.-P. M. performed the FDTD simulations. M.N., J.-P.M. and A.J.M. did the numerical modelling of the finite-sized arrays. H.T.R., T.K.H. and A.I.V. fabricated the samples. All authors discussed the results. T.K.H. and P.T. wrote the manuscript together with all authors.

## Additional information

**Competing financial interests:** The authors declare no competing financial interests.

