## [Peer Review File · Nature Communications]

Reviewers' comments:

Reviewer #1 (Remarks to the Author):

In this manuscript, the authors report on the first experimental demonstration of lasing action assisted by the surface lattice resonances in the visible regime. Specifically, the authors show experimentally how a finite periodic array of silver nanoparticles embedded in a gain medium formed by optically-pumped dye molecules can display lasing action based on both the bright and dark extended plasmonic modes supported by the structure (in this context, "bright" and "dark" label the two extended plasmonic modes that appear close to the band edge of the system). In the infinitely periodic case, due to symmetry reasons, the dark mode cannot be excited from the far field. This work is based on the fact that in a finite periodic structure, the Q-factor of the dark mode becomes finite and, therefore, it can couple to the far-field and emit laser light (in fact, more efficiently than the bright mode, due to its superior light-confinement properties). The authors present a series of measurements (such as real space radiation profile or the spacing of the resulting beating patterns) that seem to support the physical explanation they provide for their findings. In addition, the authors offer further support for the physical explanation of their findings using linear FDTD calculations.

In my view, the results reported in this paper are important and they could certainly be of interest for the broad readership of Nature Communications: I consider the present work as the next step in the series of breakthrough papers on lasing action in plasmonic system that have appeared in the last few years in top journals. The paper is also well written and organized.

However, given said that, I have the following important caveat on this manuscript. As said, the physical explanation of the results provided by the authors seems convincing. However, the large degree of optical nonlinearity present in this class of systems can lead to a number of unexpected phenomena (such mode competition, excitation of the dark mode by the bright mode via gain, etc) that cannot be captured by the linear analysis of the problem made by the authors in this work. The current state of the art of numerical simulations in nanophotonics allows simulating this class of plasmonic laser systems almost without any approximation (apart of course from the energy level system assumed for the dye molecules and the corresponding light-matter interaction parameters). To avoid any surprises down the road, I suggest the authors to consider carefully the application of those numerical methods to the structure analyzed in this work. That, in my view, would make this work much stronger and would certainly make the case clearer for the publication of this work in a top journal such as Nature Communications.

Reviewer #2 (Remarks to the Author):

This paper describes lasing from the dark and bright modes of a finite array of silver nanoparticles and dye molecules that are optically pumped. Although the premise of achieving lasing from dark modes by a coherent out-coupling mechanism seems interesting, there are several critical weaknesses that do not fully support the interpretation. The claims of dark modes for lasing require more rigorous evidence and discussion from theoretical calculations. Additionally, more information is needed to appreciate the results on lasing mode quality and beam directionality. We cannot recommend publication of this manuscript in its current form.

First, the paper claims that dark modes are a result of the finite size of the array due to the coherent build-up of dipole momentum and radiation patterns. However, the authors only use infinite arrays in FDTD simulations to demonstrate the quadrupole field pattern (Fig. 1f) of the dark mode, which is insufficient to support the assumption that finite arrays will cause the gradual change of dipolar

oscillations as described in the scheme of Fig. 4. Also, the CDA method cannot represent quadrupoles on a single nanoparticle (line 174). Additional simulations of finite arrays are necessary to see edge effects on the dipolar distributions and how the nanoparticle oscillation patterns vary from the middle to the edge, which is critical to prove the out-coupling mechanism based on a finite array sizes.

Second, the paper is lacking information for the following claims: (1) The results of control experiments on isolated PVA layers (line 75) and removing particles (line 80) are missing in the paper. (2) The paper did not show the lasing spectra for different array sizes (Fig. 5a), which is important to demonstrate the lasing quality. (3) Simulated emission spectra are missing that can differentiate between bright and dark modes. (4) There is no evidence for the beam directionality claimed in the paper (Page 4, line 127). The divergence of the beam needs to be characterized for finite arrays with different sizes.

Other issues that need to be addressed:

1. It is not clear how the emission diagram (Fig. 2a-c) is obtained from the experiment set-up (Fig. 1a). The process of $k//$ calculation and how the real-space data is transfer to the dispersion diagram should be described in Methods.
2. The paper should include more discussion over how the dark mode lasing at off-angles relate to a finite array size (Fig. 2c). The additional in-plane wave vector matching dark-mode lasing to free space should be quantified besides the empirical interpretations (line 103-108).
3. The reason for lower lasing thresholds for dark modes is not clear. Although the subradiant character of dark modes could lead to lower loss, the local field enhancement is still much weaker than that of the bright mode (Fig. 1e,f). More studies are needed to demonstrate the high quality of the dark mode such as based on quantification of mode loss by scattering and absorption in simulations.
4. The calculated bright and dark mode energy (Fig. 1 e,f) deviates from the measured lasing energy (Fig. 2d). The authors should discuss and explain this discrepancy.
5. Fig. 3a is not consistent with Fig. 3c especially with the dark stripe on the top. Also, the scale bars are missing. Higher-quality images of Fig 3a,b are necessary, and more discussion on what has been demonstrated in Fig. 3a,b is needed.
6. The paper did not include the influence of dye concentration on bright and dark mode excitation. Is there any specific reason why a high dye concentration of 31 mM was used (line 145)? Also, will high concentration of dye molecules cause intermolecular coupling that will affect the emission of dye and lasing action?

Reviewers' comments:

Reviewer #1 (Remarks to the Author):

In this manuscript, the authors report on the first experimental demonstration of lasing action assisted by the surface lattice resonances in the visible regime. Specifically, the authors show experimentally how a finite periodic array of silver nanoparticles embedded in a gain medium formed by optically-pumped dye molecules can display lasing action based on both the bright and dark extended plasmonic modes supported by the structure (in this context, "bright" and "dark" label the two extended plasmonic modes that appear close to the band edge of the system). In the infinitely periodic case, due to symmetry reasons, the dark mode cannot be excited from the far field. This work is based on the fact that in a finite periodic structure, the Q-factor of the dark mode becomes finite and, therefore, it can couple to the far-field and emit laser light (in fact, more efficiently than the bright mode, due to its superior light-confinement properties). The authors present a series of measurements (such as real space radiation profile or the spacing of the resulting beating patterns) that seem to support the physical explanation they provide for their findings. In addition, the authors offer further support for the physical explanation of their findings using linear FDTD calculations.

In my view, the results reported in this paper are important and they could certainly be of interest for the broad readership of Nature Communications: I consider the present work as the next step in the series of breakthrough papers on lasing action in plasmonic system that have appeared in the last few years in top journals. The paper is also well written and organized.

However, given said that, I have the following important caveat on this manuscript. As said, the physical explanation of the results provided by the authors seems convincing. However, the large degree of optical nonlinearity present in this class of systems can lead to a number of unexpected phenomena (such mode competition, excitation of the dark mode by the bright mode via gain, etc.) that cannot be captured by the linear analysis of the problem made by the authors in this work. The current state of the art of numerical simulations in nanophotonics allows simulating this class of plasmonic laser systems almost without any approximation (apart of course from the energy level system assumed for the dye molecules and the corresponding light-matter interaction parameters). To avoid any surprises down the road, I suggest the authors to consider carefully the application of those numerical methods to the structure analysed in this work. That, in my view, would make this work much stronger and would certainly make the case clearer for the publication of this work in a top journal such as Nature Communications.

Answer

We thank Reviewer #1 for considering our results as important, part of a series of breakthroughs in lasing action in plasmonic systems, and of interest to the broad readership of Nature Communications. We agree that providing the type of simulations mentioned by the reviewer makes the argument stronger. We have now added such state-of-the-art simulations to the manuscript. The simulations describe the electric field profiles in the system without approximation, and the gain medium is modelled as a four-level system which should describe the molecules rather well. We used finite difference time domain method (FDTD) available in commercial FDTD Solutions software by Lumerical. For the results of the simulations, see the new data in Figure 2, Methods and the related discussion. A crucial result is that both dark and bright mode lasing can be obtained from the model. Another key outcome is that the dark mode lasing threshold is lower (and intensity higher) than that of the bright mode, exactly like in our experimental observations. Quantitatively, the thresholds in the simulations are lower than in the experiments, which is expected since the real

sample and the experiment are likely to contain imperfections and loss channels not taken into account by the model. The important information gained from the simulations is: 1) we can directly identify the lasing modes with the bright and the dark modes with distinct field profiles, 2) both dark and bright modes can lase, also simultaneously, 3) the dark mode lasing has a lower threshold than the bright mode. These results are all in agreement with the experiments and strongly support our conclusions.

In addition to this, we made several improvements suggested by Reviewer #2. In particular we would like to point out that the paper now includes simulations of the finite size arrays (without the gain medium) where quadrupole, not only dipole, modes of the particles are included. These simulations (see Figure 3) demonstrate the fact that is underlying the appearance of the even beating pattern in the dark mode lasing case, namely that the light emitted from the two edges of the array in the dark mode case have opposite phases. For the sake of completeness, also the amplitude and phase distributions of the quadrupolar excitations are now included in the Supplemental Information. With these two (lasing with FDTD, and the finite array simulations including quadrupole moments) important simulation results added, together with other improvements, we hope the reviewer will find the manuscript suited for publication in Nature Communications.

Reviewer #2 (Remarks to the Author):

This paper describes lasing from the dark and bright modes of a finite array of silver nanoparticles and dye molecules that are optically pumped. Although the premise of achieving lasing from dark modes by a coherent out-coupling mechanism seems interesting, there are several critical weaknesses that do not fully support the interpretation. The claims of dark modes for lasing require more rigorous evidence and discussion from theoretical calculations. Additionally, more information is needed to appreciate the results on lasing mode quality and beam directionality. We cannot recommend publication of this manuscript in its current form.

****Answer****

We thank the Reviewer #2 for stating that our results are interesting, and for pointing out a detailed list of improvements that are needed to fully confirm the findings. We have now done substantial effort to provide the necessary evidence and theoretical calculations, as described in detail below.

First, the paper claims that dark modes are a result of the finite size of the array due to the coherent build-up of dipole momentum and radiation patterns. However, the authors only use infinite arrays in FDTD simulations to demonstrate the quadrupole field pattern (Fig. 1f) of the dark mode, which is insufficient to support the assumption that finite arrays will cause the gradual change of dipolar oscillations as described in the scheme of Fig. 4. Also, the CDA method cannot represent quadrupoles on a single nanoparticle (line 174). Additional simulations of finite arrays are necessary to see edge effects on the dipolar distributions and how the nanoparticle oscillation patterns vary from the middle to the edge, which is critical to prove the out-coupling mechanism based on a finite array sizes.

****Answer****

We now present simulations that elucidate this critical issue pointed out by the referee. We used the T-matrix multiple scattering method, which can account for both dipolar and quadrupolar excitations in the nanoparticles for finite-sized arrays, to demonstrate that indeed the distributions vary from the centre of the array towards the edges as argued in the original manuscript. These results are presented in Figure 3, Methods and Supplementary Information.

Second, the paper is lacking information for the following claims: (1) The results of control experiments on isolated PVA layers (line 75) and removing particles (line 80) are missing in the paper. (2) The paper did not show the lasing spectra for different array sizes (Fig. 5a), which is important to demonstrate the lasing quality. (3) Simulated emission spectra are missing that can differentiate between bright and dark modes. (4) There is no evidence for the beam directionality claimed in the paper (Page 4, line 127). The divergence of the beam needs to be characterized for finite arrays with different sizes.

****Answer****

- (1) We have now added specific information about the particle removal and PVA spacer layer experiments in the Supplementary Information, see Figs S1-S4 and the related discussion.
- (2) We have now added the new Figure 5 c, where lasing spectra for different array sizes are shown. The new figure is discussed in the manuscript together with other array size dependent data of Fig. 5.
- (3) Also Reviewer #1 encouraged us to add state-of-the-art simulations where the full field profiles are combined with a four-level gain medium. Importantly, such simulations are able to differentiate between bright and dark modes. We have now replaced the results obtained from the neoclassical lasing model with finite difference time domain simulations, see inset of Fig. 2 e, Methods and Supplementary Information.
- (4) We have now added a description on how the beam directionality can be determined from the momentum (k) distribution of the emitted light in the Methods section, and given specific numbers for the beam directionality (it is typically 0.3 degrees), also for different array sizes, see Fig 5 c. We now mention specific numbers for the beam directionality at the place the Reviewer mentions (Page 4, line 127 of the original manuscript).

Other issues that need to be addressed:

1. It is not clear how the emission diagram (Fig. 2a-c) is obtained from the experiment set-up (Fig. 1a). The process of $k//$ calculation and how the real-space data is transfer to the dispersion diagram should be described in Methods.

****Answer****

The emission and dispersion diagrams are obtained by Fourier plane imaging, thus no conversion from real space data is needed. The real space data is obtained separately. Indeed the Fourier plane imaging was not explained in sufficient detail in the original manuscript, but now we have explained how the data in Fig.2 are obtained. Moreover, we now describe in more detail how $k//$ relates to the angle of the emitted/transmitted light and how that is seen at the spectrometer in the Fourier plane measurement. These new explanations can be found in the Methods section.

2. The paper should include more discussion over how the dark mode lasing at off-angles relate to a finite array size (Fig. 2c). The additional in-plane wave vector matching dark-mode lasing to free space should be quantified besides the empirical interpretations (line 103-108).

****Answer****

The dark mode in our system couples out due to specific behaviour of the nanoparticle oscillations when the dark mode is excited: in the middle of the array, the particles present quadrupole oscillations, while at the edges the oscillations also have a dipolar component and therefore can radiate to the free space. Importantly, however, the phase of the dipolar oscillation is the opposite in the two edges, thus also the light leaking out has opposite phase on the two edges. In the original manuscript, this argument was illustrated by the schematic Figure 4. In the revised manuscript, we have performed actual simulations which quantitatively show that indeed for the dark mode the

quadrupoles dominate in the centre while the dipoles maximize near the edges, and show opposite phases, see the new Figure 3 and Supplementary Information; the opposite phases in particular are shown in Figure 3 f. Our array can then be understood somewhat analogous to a double slit, with the edges representing the slits. Light that is emitted on a spatially restricted area such as edge or slit obtains a wide range of momenta. However, since there are two slits, interference affects the angles/momenta of the light visible in the far field. In case the phase of the light at the two edges is the same, an interference maximum occurs at the centre, corresponding to $k=0$, analogous to a usual double slit. In contrast, when the phases are opposite, destructive interference will occur at the centre, and constructive interference produces peaks at off angle, as seen in our case. According to this picture, the separation between these peaks should be related to the distance between the light sources, i.e. the edge positions. This is indeed the case as shown quantitatively by experiments in Figure 5. The new Figures 3 c-f and 5 c combined with the original Figure 5 a,b now give convincing evidence that this picture captures the essential features of the outcoupling mechanism. We also rewrote the last two sentences of the caption of Figure 4 to be more instructive.

3. The reason for lower lasing thresholds for dark modes is not clear. Although the subradiant character of dark modes could lead to lower loss, the local field enhancement is still much weaker than that of the bright mode (Fig. 1e,f). More studies are needed to demonstrate the high quality of the dark mode such as based on quantification of mode loss by scattering and absorption in simulations.

***** Answer*****

Indeed such simulations are crucial and were asked for also by Reviewer #1. The essential results have now been added to the manuscript, see inset of 2 e. The simulations describe the electric field profiles in the system without approximation, and the gain medium is modelled as a four-level system which should describe the molecules rather well. We used finite difference time domain method (FDTD) available in commercial FDTD Solutions software by Lumerical. A crucial result is that the dark mode lasing threshold is lower (and intensity higher) than that of the bright mode, just like in our experimental observations.

4. The calculated bright and dark mode energy (Fig. 1 e,f) deviates from the measured lasing energy (Fig. 2d). The authors should discuss and explain this discrepancy.

***** Answer*****

This minute (a few nanometers) difference is likely due to small differences in the values of the refractive index in the simulations and the experiments. A fourth digit change of the refractive index leads already to a few nanometer change, for instance $375 \text{ nm} \times 1.52 = 570 \text{ nm}$, while $375 \text{ nm} \times 1.528 = 573 \text{ nm}$. To clarify this, we have added in the beginning part of the paragraph that starts “We combine the lattices with...” the following sentence: “Addition of the dye molecules modifies the refractive index surrounding the structure, shifting the SLR resonances slightly from Fig.1”.

5. Fig. 3a is not consistent with Fig. 3c especially with the dark stripe on the top. Also, the scale bars are missing. Higher-quality images of Fig 3a,b are necessary, and more discussion on what has been demonstrated in Fig. 3a,b is needed.

***** Answer*****

Scale bars have been added, and higher quality images for Figs. 3 a,b have been provided. Discussion on how the real space measurements and dipole distributions give information about whether the dark or the bright mode is lasing, and about the outcoupling mechanism, have been added in connection with the new Figures 3 c-f. The old figure, including only extinction cross section calculated from dipolar moments, have now been replaced with extended calculations

including quadrupolar excitations as well. Figs. 3 c and d now show the dipole moment distributions at both dark and bright mode energies. As the square of the induced dipole moment of the particle is proportional to the radiation intensity, the reader can now make a qualitative connection between the measured real space images in Figs 3 a,b and the calculated dipole moment distributions shown in 3 c,d. Further, we included the calculated phase distribution of the dipole moments in order clarify the origin of the even beating pattern of the dark mode. For the sake of completeness, also the amplitude and phase of the quadrupolar excitations are included in the Supplemental Information. Altogether, these simulations are in full agreement with the original claims, namely: 1) for the bright mode, the dipole moments maximize at the centre and are in phase, 2) for the dark mode the quadrupolar excitation at the centre of the array gradually evolves into a dipolar excitation (with a pi phase shift) towards the edges.

6. The paper did not include the influence of dye concentration on bright and dark mode excitation. Is there any specific reason why a high dye concentration of 31 mM was used (line 145)? Also, will high concentration of dye molecules cause intermolecular coupling that will affect the emission of dye and lasing action?

*****Answer*****

Studying the influence of dye concentration is definitely an interesting issue, but beyond the scope of this work. There was no specific reason for choosing exactly 31 mM concentration, but the order of magnitude was chosen because we have prior experience on about 1 – 300 mM concentrations, and for high concentrations of hundreds of mM aggregation of the molecules starts to play a role, as the reviewer suggests. On the other hand, we have been typically unsuccessful in reaching lasing for very small concentrations. Therefore, about 30 mM is a good “intermediate” value. Systematic studies of the effect of concentration are definitely an interesting topic for future research. We have added to the manuscript the comment: “For concentrations of this order of magnitude, formation of aggregates is not significant” and a reference to our earlier work where concentrations that typically lead to aggregation of the molecules have been quantified.

We thank the Reviewer once more for thorough reading of the manuscript and the comments that have helped substantially improve the manuscript. We hope the manuscript is now ready for publication in Nature Communications.

Reviewers' comments:

Reviewer #1 (Remarks to the Author):

The authors have successfully addressed all my comments and suggestions. I do not have further comments or suggestions on this work.

Reviewer #2 (Remarks to the Author):

The authors have done a very nice job addressing most of our major concerns; however, more clarification is still needed regarding language and mechanism so that the paper can be best appreciated. Other minor revisions are necessary.

1. Hybrid dark mode in a finite array. The authors should clarify the origin of the dark mode in a finite array and its relationship with a quadrupolar mode distribution. Usually, the dark mode occurs at zero wavevector (Fig. 1d) in an infinite array with quadrupolar mode distribution for all NPs (Fig. 1f). But in this case, the finite size is modifying the overall field distribution of NPs such that the "dark" mode seems to be more of a hybrid mode of a true dark mode and finite size confinement. And, this mode seems to be visible in the far-field at non-zero wavevectors. Figure 5a suggests that the k-vector spacing for the dark mode is modified by the patch size. Also, from Fig. 5c, the "dark" lasing mode wavelength red-shifted with increased patch size, which suggests a hybrid mechanism. Also, the dark mode here specifically refers to one band-edge mode missing in the far-field in plasmonic lattices. There are other categories of dark modes in plasmonics system to present a subradiant mode, especially in metamaterials with some symmetry-breaking features [Phys. Rev. Lett. 101, 047401 (2008); Applied Physics Letters 100, 131101 (2012)]. The authors can incorporate this discussion to the current manuscript.

2. Quadrupolar mode distribution. The quadrupolar mode distribution of this hybrid dark mode should be better explained. In the manuscript, evidence of the quadrupole mode distribution is based on the simulation of infinite arrays (Fig. 1f), but in the Fig. 4 scheme, it seems that the "dark" mode has an overall phase tuning across the patch, while the quadrupolar mode is only the single one in the middle.

3. Simulations of lasing action. Lumerical was used for simulations, and bright/dark mode lasing is plotted separately at specific monitor locations (Figure S6 caption). We think the authors should address the issue of why the lasing intensity of the bright mode is weaker than the pump in Fig. S6a, since usually, if it's weaker, lasing action hasn't built up yet.

4. Dark mode field distribution. The authors used the T-matrix method to include both dipole and quadrupolar modes to reproduce the dipole momentum pattern. However, this is not a direct way to prove the phase shift scheme in Fig. 4 compared to depicting directly near-field distribution or phase change around NPs. In Fig. 3b, the dipole moments at the dark lasing mode is plotted instead of quadrupole moments to match the real-space pattern. We find it counter-intuitive to correlate the real-space lasing pattern at dark mode with the dipole moment distribution directly.

Response to reviewers comments

Reviewer #1 (Remarks to the Author):

The authors have successfully addressed all my comments and suggestions. I do not have further comments or suggestions on this work.

Reviewer #2 (Remarks to the Author):

The authors have done a very nice job addressing most of our major concerns; however, more clarification is still needed regarding language and mechanism so that the paper can be best appreciated. Other minor revisions are necessary.

1. Hybrid dark mode in a finite array. The authors should clarify the origin of the dark mode in a finite array and its relationship with a quadrupolar mode distribution. Usually, the dark mode occurs at zero wavevector (Fig. 1d) in an infinite array with quadrupolar mode distribution for all NPs (Fig. 1f). But in this case, the finite size is modifying the overall field distribution of NPs such that the “dark” mode seems to be more of a hybrid mode of a true dark mode and finite size confinement. And, this mode seems to be visible in the far-field at non-zero wavevectors. Figure 5a suggests that the k-vector spacing for the dark mode is modified by the patch size. Also, from Fig. 5c, the “dark” lasing mode wavelength red-shifted with increased patch size, which suggests a hybrid mechanism. Also, the dark mode here specifically refers to one band-edge mode missing in the far-field in plasmonic lattices. There are other categories of dark modes in plasmonics system to present a subradiant mode, especially in metamaterials with some symmetry-breaking features [Phys. Rev. Lett. 101, 047401 (2008); Applied Physics Letters 100, 131101 (2012)]. The authors can incorporate this discussion to the current manuscript.

*** Response***

The mode in a finite array is indeed different from the one in an infinite array, and it is important to discuss this further in the manuscript. We have added the following clarifying sentence in the text (in the paragraph starting “We hypothesize...”): “The dark mode in a finite lattice is thus a hybrid mode containing also features typical for a bright mode of an infinite lattice, such as dipoles, which makes it visible in the far field.” We have also added a similar sentence in the end of the caption of Figure 4: “The dark mode in a finite lattice is thus different from the one in an infinite lattice: it features dipole moments and can thus be viewed as a hybrid mode with characteristics of both the bright and dark modes of an infinite lattice.” The comment of the red-shift by the reviewer is very interesting, and supports the hybrid nature. However, we view this is a subject of further study and we decided not to comment the red-shift in this manuscript. We agree that is it useful to clarify the nature of the dark modes we are considering with respect to some other types of dark modes discussed in the literature. To this end, we have added the following sentence, as suggested by the reviewer, in the end of the paragraph “We fabricate...”: “A dark mode here specifically refers to one band-edge mode missing in the far field. There are other categories of dark modes in plasmonics systems which present a subradiant mode, especially in metamaterials with symmetry-breaking features [Phys. Rev. Lett. 101, 047401 (2008); Applied Physics Letters 100, 131101 (2012)].” We have added the two references mentioned by the reviewer.

2. Quadrupolar mode distribution. The quadrupolar mode distribution of this hybrid dark mode should be better explained. In the manuscript, evidence of the quadrupole mode distribution is based on the simulation of infinite arrays (Fig. 1f), but in the Fig. 4 scheme, it seems that the “dark” mode has an overall phase tuning across the patch, while the quadrupolar mode is only the single one in the middle.

*** Response***

True, the caption of Figure 4 was not ideally clear: we did not explain that quadrupoles exist also elsewhere than right in the middle. The purpose of the Figure 4 was to explain how the dipoles build up, but one should not forget the quadrupoles. We have now modified the figure caption accordingly. The modified sentences read: “For a dark mode, particles in the array are driven by left- and right-propagating radiation fields, which destructively interfere at the particle locations and therefore create a standing wave node and a quadrupole excitation.” and, after a few sentences “This results in a buildup of dipole moments whose relative weight compared to the quadrupoles gradually increases when moving away from the center of the array.”

3. Simulations of lasing action. Lumerical was used for simulations, and bright/dark mode lasing is plotted separately at specific monitor locations (Figure S6 caption). We think the authors should address the issue of why the lasing intensity of the bright mode is weaker than the pump in Fig. S6a, since usually, if it's weaker, lasing action hasn't built up yet.

*** Response***

The pump power that the picture corresponds to is well above the lasing threshold for both modes. A large fraction of the pump energy goes to the dark mode, whose output power is about an order of magnitude bigger than that of the bright mode, consequently the pump peak is higher compared to the bright mode than in the dark mode case. We have now added to the figure caption of Figure S6 the following sentence to clarify that both modes are lasing and that most of the pump power is channeled to the dark mode: “For this pump power, both modes are well above the lasing threshold, see Fig. 2 g of the main text, but the dark mode output power is about an order of magnitude higher than that of the bright mode.”

4. Dark mode field distribution. The authors used the T-matrix method to include both dipole and quadrupolar modes to reproduce the dipole momentum pattern. However, this is not a direct way to prove the phase shift scheme in Fig. 4 compared to depicting directly near-field distribution or phase change around NPs. In Fig. 3b, the dipole moments at the dark lasing mode is plotted instead of quadrupole moments to match the real-space pattern. We find it counter-intuitive to correlate the real-space lasing pattern at dark mode with the dipole moment distribution directly.

*** Response***

The actual field distributions at the sample (i.e. the near field) have important contributions from the quadrupoles and, true, the lasing fields in the near field could not be related to the dipole moments only. However, what we observe in the real-space images is the far field radiation emitted

from the sample. This is dominated by dipoles. We have now explained this better in the manuscript by adding to the paragraph starting “From Figs. 2 d-f...” the modified text: “A striking difference between the two cases is observed also in real space images of the lasing action. These far field microscope images correspond to the real space distribution of radiation from the sample but are not the same as the near-field distribution.” We also added there the modified text: “The real-space images in far field result from radiation emitted by the sample. The far field emission power is dominated by dipoles, thus the radiated far field intensities in Figs. 3 a-b are proportional to the square of induced dipole moments.”

REVIEWERS' COMMENTS:

Reviewer #2 (Remarks to the Author):

The authors have clarified important points, and I am happy to recommend publication of this manuscript.